# An Improved Genetic Algorithm for Emergency Decision Making under Resource Constraints Based on Prospect Theory

**Leiwen Chen \*, Yingming Wang and Geng Guo**

School of Economics & Management, Fuzhou University, Fuzhou 350116, China; ymwang@fzu.edu.cn (Y.W.);
N160720099@fzu.edu.cn (G.G.)
**\*** Correspondence: cleiwen114@gmail.com; Tel.: +86-180-6084-8506

**Abstract:** The study of emergency decision making (EDM) is helpful to reduce the difficulty of decision making and improve the efficiency of decision makers (DMs). The purpose of this paper is to propose an innovative genetic algorithm for emergency decision making under resource constraints. Firstly, this paper analyzes the emergency situation under resource constraints, and then, according to the prospect theory (PT), we further propose an improved value measurement function and an emergency loss levels weighting algorithm. Secondly, we assign weights for all emergency locations using the best–worst method (BWM). Then, an improved genetic algorithm (GA) based on prospect theory (PT) is established to solve the problem of emergency resource allocation between multiple emergency locations under resource constraints. Finally, the analyses of example show that the algorithm can shorten the decision-making time and provide a better decision scheme, which has certain practical significance.

**Keywords:** genetic algorithms; prospect theory; emergency decision making; resource constraints

## 1. Introduction

An emergency ordinarily leads to great casualties and enormous economic losses, such as September 11 attacks in 2001, Wenchuan earthquake in 2008, West African Ebola virus epidemic in 2014, and Typhoon Mangkhut in 2018 (the situation of casualties and enormous economic losses can be seen in Table 1). The topic of emergency response has attracted some scholars' attention recently [1–3]. How to deal with emergencies in time has become an urgent problem to be solved by decision makers (DMs). Especially in the situation of resource constraints and imbalance of supply and demand, it is of practical significance to study how to reasonably allocate and dispatch emergency resources and make feasible emergency schemes for shortening the time of emergency decision making and improving the working efficiency of the emergency department.

**Table 1.** The situation of casualties and enormous economic losses.

| Emergency Events | September 11 Attacks | Wenchuan Earthquake | West African Ebola Virus Epidemic | Typhoon Mangkhut |
| --- | --- | --- | --- | --- |
| Casualties | 2996 | 87,587 | 11,310 | 134 |
| economic losses | $5 trillion | $150 billion | $3.8–32.6 billion | $3.74 billion |

[1] The data come from the literature [4–7].

Nowadays, the study of emergency decision making (EDM) is partly based on group decision-making problems with different criteria weighting methods, such as the distance-based

method and the entropy method [8,9]. Some scholars considered the data in EDM as being in the form of fuzzy sets and constructed fuzzy preference relations and fuzzy aggregation operators [10–13] in order to describe the decision-making data more accurately. In terms of the alternatives ranking problems, scholars usually employ the objective method to make a comparison, such as technique for order preference by similarity to an ideal solution (TOPSIS) and Visekriterijumska Optimizacija I Kompromisno Rješenje (VIKOR) [14–17], which enriched the solution of EDM problems.

Although the above studies can deal with EDM problems to some extent, they can not satisfy the requirement of quick decision making in EDM. Further, they rarely consider the problem of emergency resource allocation under resource constraints.

The existing research on emergency resource allocation mainly focuses on the linear programming algorithm. Ge et al. [18] set up the main objective (the least number of victims) and the secondary objective (the least total transportation cost), and established the optimizing model by the means of linear programming. According to the Markov property of the material needs among different locations in typhoon days, Zhan et al. [19] proposed an optimal scheme selection model based on binary particle swarm optimization algorithm (BPSO). Taskin et al. [20] previously proposed a number prediction model of typhoon based on historical data, and then on this basis, proposed an emergency resources inventory controlling strategy.

The above research algorithms have strong objectivity, and the subjective consideration of value measurement and weighting algorithm is less. In addition, the emergency situation of the above algorithm depends more on the probability setting, but it is difficult to obtain the data in the practical application. On this basis, a value measurement function and an emergency loss level weighting algorithm based on prospect theory are proposed. On this basis, the value measure function and weighted algorithm are combined with a genetic algorithm to solve the problem of emergency resource allocation between multiple emergency locations with fuzzy requirements under resource constraints.

## 2. Preliminaries

In this section, some associated definitions on prospect theory and best–worst method (BWM) are reviewed, and the genetic algorithm is also discussed.

### 2.1. Prospect Theory

Because traditional expected utility theory can not explain the irrational behavior under risk and uncertainty, which leads to the deviation of the forecast decision result, prospect theory (PT) is proposed [21]. The core idea of PT is that people's judgments of gains and losses are usually based on reference points, and DMs are often "risk averse" when facing risks.

DMs first set reference points. According to prospect theory [21], the value function can be represented by

$$v(\Delta x) = \begin{cases} (\Delta x)^{\alpha}, \Delta x \geq 0 \\ -\theta(-\Delta x)^{\beta}, \Delta x < 0 \end{cases}, \tag{1}$$

where $\Delta x$ is the difference between the actual value and the reference point, which is used to describe the gains and losses of decision-makers. For $\Delta x \geq 0$, it denotes that DMs get more than they expected; for $\Delta x \leq 0$, it denotes that DMs get less than they expected. $\alpha$ and $\beta$ are exponent parameters ($0 < \alpha$, $\beta < 1$); $\theta$ is the loss aversion parameter ($\theta > 1$), which indicates that DMs are more sensitive to losses than gains; and $\alpha = 0.89$, $\beta = 0.92$, and $\theta = 2.25$.

In order to take humans' psychological behavior into account, some scholars have applied PT to decision-making problems. Liu et al. [22] used cumulative prospect theory to analyze the decision-makers' psychology and calculated the prospect value of each response result using the cumulative negative effect value of the response result and the occurrence weight of each scenario. In addition, Wang et al. [23] used PT to explain DMs' bounded rationality when facing emergencies

under risk and uncertainty. Liu et al. [24] combined PT with interval probability in order to develop a novel intelligent optimization algorithm.

*2.2. Best–Worst Method*

The best–worst Method (BWM) was improved based on AHP by Rezaei [25] in 2015, changing the pairwise comparison from AHP into the comparison between the remaining criteria and the best–worst criteria. BWM has been used to rank criteria by some scholars. Rezaei et al. [26] presented a supplier selection life cycle approach, which integrated qualitative, quantitative, traditional business, and environmental criteria using BWM. Gupta et al. [27] used BWM to rank the criteria of green innovation for supplier selection. In addition, Amoozad Mahdiraji et al. [28] applied BWM in order to determine the weights and priority of the identified criteria.

Firstly, the best important and least important criteria are identified by the DMs. Then, the preference of the best and the worst criterion over all the other criteria is determined using a number between 1 and 9. Let $A_B$ ($A_W$) be the sets of comparison values between the best (least) important criteria and the other criteria. The vectors would be $A_B = (a_{B1}, a_{B2}, \cdots, a_{Bn})$ and $A_W = (a_{1W}, a_{2W}, \cdots, a_{nW})$. It is clear that $a_{BB} = a_{WW} = 1$. The calculating function is as follows:

$$a_{ij} = a_{ik} * a_{kj}, \text{ for all } k \neq i, j, \tag{2}$$

$$a_{ji} = \frac{1}{a_{ij}}. \tag{3}$$

Finally, according to the vectors, we can get the best–others and the worst–others comparative matrices. The optimal and worst comparison matrices are standardized by Equation (4).

$$\overline{A} = \sqrt{A_B * A_W} \tag{4}$$

In this paper, this method was employed in order to calculate the priority weighting of the areas. The advantages of the BWM are the following:

(1)   Pairwise comparison needs to do $n^2$ comparisons, and the BWM only needs $2n$ comparisons, which greatly simplifies the weighting steps.
(2)   By reducing the amount of data, the results are consistent and reliable.

*2.3. Genetic Algorithm*

Genetic algorithm (GA) is a computational model that simulates Darwin's natural evolutionary process. It is a method to find the optimal solution by simulating the natural evolution process. In the first part, the initial population is generated by binary coding, and the individual is selected according to the fitness value of the problem domain. After that, genetic operators of natural genetics are used to hybridize and mutate the original population to produce the next generation population representing the new solution set. By repeating the above steps, the candidate population of the scheme set can be closer to the optimal solution. Finally, the population is the approximate solution of the optimal solution.

The advantages of the genetic algorithm are as follows:

(1)   Multiple individuals can be processed at the same time, that is, multiple solutions in the search space can be evaluated simultaneously, avoiding the risk of being trapped in local optimum.
(2)   Candidates are selected from the overall situation according to fitness value. It has a wide coverage, which helps to improve the accuracy of the results.
(3)   The fitness function and its domain can be adjusted according to the specific situation, which greatly expands its application range.

Some scholars have applied GA to decision-making problems. For example, Zhu et al. [29] proposed an improved GA for a resource-constrained project scheduling problem. A multi-objective genetic algorithm was employed by Yang et al. [30] to find the set of Pareto solutions to the optimization model of ship form evaluation. Tan et al. [31] combined GA with hesitant intuitionistic fuzzy sets, in order to get the optimal solution of decision making. However, the methods in literature above were all objective methods, which neglected the DMs' bounded rationality. In the proposed approach of this paper, GA was combined with PT in order to take the DMs' subjective factors into account.

## 3. An Algorithm for EDM with GA

### 3.1. Description of EDM Problems with GA

When there are $m$ places where emergencies occur at the same time, DMs usually can not get the specific situations of the emergencies in the first place. The decision making at this time needs to be timely and effective. Let C = {$C_1$, $C_2$, $C_3$, ... , $C_n$} be a set of emergency loss levels. Set the casualty situation as D = {$D_1$, $D_2$, $D_3$, ... , $D_n$}. $D_i$ indicates the casualty situation when the emergency loss level $C_i$ happens, which is expressed by the interval number as $[d_i^L, d_i^H]$. Let E = {$E_1$, $E_2$, $E_3$, ... , $E_n$} be a set of the property loss situation, and $E_i$ indicates the property loss situation when the emergency loss level $C_i$ happens, which is expressed by the interval number as $[e_i^L, e_i^H]$.

### 3.2. An Algorithm for Determining the Weights of Emergency Loss Levels

When the level of emergency loss is low, the effect of each emergency scheme is not different, so it is difficult for DMs to distinguish which scheme is the best from the beginning. With the further evolution of the emergency situation from low to high, the scheme with good effect under the low emergency loss level may not be able to adapt to the high emergency loss level. When emergencies get serious, DMs also wants to play a bigger role in decision making. Therefore, DMs should give a higher level of emergency weight in decision making.

Generally, losses happen when an emergency occurs, that is, the DM's gain-and-loss interval is $[-\infty, 0]$, so the decision-making reference point is set to 0. According to prospect theory, the psychological value measurement equations of casualty and property losses when facing emergencies are defined as follows:

$$v(D_i) = \frac{[-\theta(d_i^L)^\beta] + [-\theta(d_i^H)^\beta]}{2}, \text{ for all } d_i \geq 0, \ i \in [0, n], \tag{5a}$$

$$v(E_i) = \frac{[-\theta(e_i^L)^\beta] + [-\theta(e_i^H)^\beta]}{2}, \text{ for all } e_i \geq 0, \ i \in [0, n]. \tag{5b}$$

In order to eliminate the influence of dimensions, the above values are normalized as follows:

$$\overline{v(D_i)} = \frac{v(D_i)}{|v(D_i)|_{\max}}, \text{ for all } |v(D_i)|_{\max} = \max\{v(D_1), v(D_2), \ldots, v\}, \tag{6}$$

$$\overline{v(E_i)} = \frac{v(E_i)}{|v(E_i)|_{\max}}, \text{ for all } |v(E_i)|_{\max} = \max\{v(E_1), v(E_2), \ldots, v(E_n)\}. \tag{7}$$

Combining $\overline{v(D_i)}$ with $\overline{v(E_i)}$, the comprehensive evaluation value of the emergency loss levels can be obtained by the above equations:

$$v(C_i) = \overline{v(D_i)} + \overline{v(E_i)}. \tag{8}$$

The weighting coefficient equation of the emergency loss level can be obtained as follows:

$$\omega_i = \frac{v(C_i)}{\sum_{i=1}^{n} v(C_i)} \omega_j, \text{ for all } j = 1, 2, \ldots, \text{m},$$

(9)

where $\omega_j$ represents the importance of the emergencies area, which is obtained by the BWM.

### 3.3. Construction of Emergency Decision-Making GA Based on Resource Constraints

Emergency decision making has higher requirements for decision-making time and needs to be able to complete the decision-making process quickly and effectively. There are many decision variables involved in this paper, and the solution processes are complicated.

The traditional decision method is time-consuming and laborious, and it is difficult to get an accurate answer. Genetic algorithm avoids complex solving operations and selects individuals to form new populations according to fitness function. Genetic operators are used for cross and mutation combinations to generate new solution populations. This process is called population evolution. Through the evolution from generation to generation, the new generation population has better environmental adaptability than the old generation. After many cycles of evolution, we can get some approximate optimal solutions of the problem.

(1)  Design of the Fitness Function

In different regions, demands for relief workers and materials should be different at all emergency loss levels. Relief workers and materials both belong to emergency resources, and the demands of emergency resources are uncertain. Therefore, the demands can be described in the form of intervals. In this paper, the fitness function is the number of relief workers and the comprehensive evaluation value of planned dispatch materials.

The number of relief workers demanded in a place $j$ when the emergency loss level is $i$ is denoted by $X_i = [\underline{x}_i, \overline{x}_i]$, where $\underline{x}_i$ is the lower bound of the demand, and $\overline{x}_i$ is the upper bound of the demand. $x_i$ denotes the number of rescuers scheduled to be sent. This paper improves the value function of PT according to the situation of EDM, and defines the evaluation value equations of $x_i$ as follows:

$$v(x_i)_j = \begin{cases} -\theta \left[ -\left( \dfrac{x_i - \underline{x}_i}{\overline{x}_i - \underline{x}_i} \right) \right]^{\beta}, & x_i < \underline{x}_i \\[3mm] \left( \dfrac{x_i - \underline{x}_i}{\overline{x}_i - \underline{x}_i} \right)^{\alpha}, & \underline{x}_i \leq x_i \leq \overline{x}_i \\[3mm] 1, & x_i > \overline{x}_i \end{cases}$$

(10)

According to Equation (10), if $x_i < \underline{x}_i$, then we say that the number of rescuers does not meet the minimum standard of emergency loss level, so the psychological expectation for the DM is negative; if $\underline{x}_i \leq x_i \leq \overline{x}_i$, then we say that the decision results are in line with the prospects; if $x_i > \overline{x}_i$, then we do not think redundant rescuers make better decisions, nor do people with diabetes increase their psychological expectations.

Through Equation (10), the total value function for sending $x_i$ relief workers to place $j$ can be defined as follows:

$$v(x)_j = \sum_{i=1}^{n} v(x_i) \omega_i.$$

(11)

Similarly, the number of materials demanded in a certain place when the emergency loss level is $i$ is denoted by $Y_i = [\underline{y}_i, \overline{y}_i]$. The evaluation value equations of $y_i$ is defined as follows:

$$
v(y_i)_j = \begin{cases} -\theta\left[-\left(\frac{y_i - \underline{y}_i}{\overline{y}_i - \underline{y}_i}\right)\right]^{\beta}, y < \underline{y}_i \\ \left(\frac{y_i - \underline{y}_i}{\overline{y}_i - \underline{y}_i}\right)^{\alpha}, \ \underline{y}_i \leq y_i \leq \overline{y}_i \\ 1, \ y_i > \overline{y}_i \end{cases} .
\tag{12}
$$

$$
v(y)_j = \sum_{i=1}^{n} v(y_i)\omega_i.
\tag{13}
$$

Combining $v(x)$ with $v(y)$, the objective function in this paper is the following:

$$
\max \sum_{j=1}^{m} [v(x)_j + v(y)_j],
\tag{14}
$$

where $[v(x)_j + v(y)_j]$ is the comprehensive evaluation value of decision.

In addition to this, we need to add the following constraints:

$$
\sum_{j=1}^{m} v(x)_j \leq S_p,
\tag{15}
$$

$$
\sum_{j=1}^{m} v(y)_j \leq S_g,
\tag{16}
$$

where $S_p$ and $S_g$ represent the total number of rescuers and materials owned by DMs, respectively.

(2)　Approach for Emergency Decision Making with Improved GA

On the basis of the above analysis, an innovative method of improving the genetic algorithm to solve the EDM problem is proposed. The specific steps are summarized below and in the flowchart depicted in Figure 1.

**Step 1.** Initialize the population and set the number of iterations for the population.

**Step 2.** Calculate the fitness function of the population through Equations (10) to (16).

**Step 3.** Select excellent individuals by the roulette method. Through crossover and mutation, the new populations are formed.

**Step 4.** Repeat steps 3 and 4 and iterate according to the number of iterations to make the data converge as much as possible to find the best value.

**Step 5.** After the iteration is completed, stop the calculation and output the optimal value.

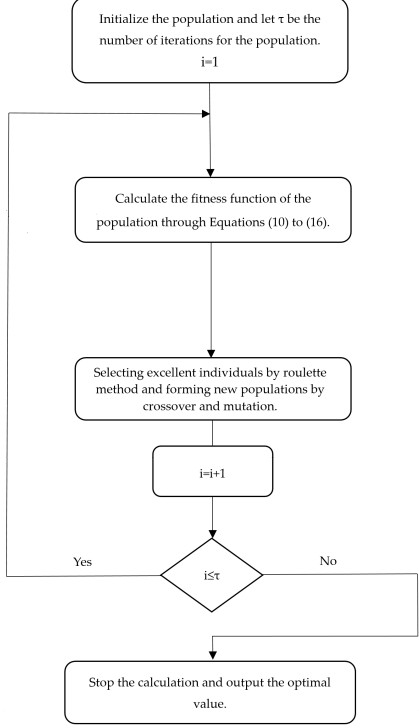

**Figure 1.** The flowchart of the entire steps for emergency decision making (EDM) with improved genetic algorithm (GA).

## 4. Practical Example for EDM with Improved GA

In this section, an emergency decision-making example is provided to show the application of the proposed method.

How to make emergency decision quickly and efficiently is an important task of the decision support system. An earthquake occurred in cities, causing serious damage to A, B, C, D, and E districts. By analyzing the severity of the emergencies, DMs classify the emergency loss level into light ($C_1$), moderate ($C_2$), and severe ($C_3$). The criteria of emergency loss levels are shown in Table 2. Each emergency loss level consists of different demand quantities of rescuers and materials, which are denoted by interval numbers. Specific requirements are shown in Table 3. At present, there are 170 rescuers and 350 kilograms of materials available for scheduling.

**Table 2.** Criteria of emergency loss levels.

|    | Casualties | Property Loss |
|----|-----------|---------------|
| C1 | [0,5]     | [0,50]        |
| C2 | [6,15]    | [50,100]      |
| C3 | [16,30]   | [100,200]     |

**Table 3.** Demand quantities of rescuers and materials.

| Area | Demand of Rescuers(X) | | | Demand of Materials(Y) | | |
|------|-------|----------|--------|-------|----------|--------|
|      | Light | Moderate | Severe | Light | Moderate | Severe |
| A | [10,15] | [15,25] | [25,30] | [20,30] | [30,40] | [40,55] |
| B | [10,20] | [20,30] | [30,40] | [30,40] | [40,50] | [50,65] |
| C | [15,20] | [20,35] | [35,50] | [40,60] | [60,80] | [80,100] |
| D | [5,15] | [15,25] | [25,35] | [45,65] | [65,90] | [90,110] |
| E | [20,30] | [30,50] | [50,60] | [45,65] | [65,90] | [90,110] |

In what follows, the proposed method in this paper is employed to solve this example.

**Step 1.** Determine the weights of emergency loss levels.

(i) Through Equations (5) to (8), the evaluation values of emergency loss levels are shown in Table 4.
(ii) Through Equation (9), the weights of emergency loss levels are calculated as follows:

$$\omega = (\omega_1, \omega_2, \omega_3) = (0.0922,\ 0.3052,\ 0.6026).$$

**Table 4.** Evaluation values of emergency loss levels.

| Value | Casualties | Property Losses |
|-------|-----------|-----------------|
| C1 | −4.9454 | −41.1344 |
| C2 | −19.4366 | −118.9654 |
| C3 | −40.1294 | −225.0962 |

**Step 2.** The area weights are determined by BWM. According to past experience, DMs determined area E as the best important area and area A as the worst important region. The judgment matrices of areas A and E are calculated through Equations (2) to (4).

(i) Determine the preference of area E over all the other areas. The resulting best-to-others vector would be the following:

$$A_B = (a_{EA}, a_{EB}, a_{EC}, a_{ED},) = (5, 3, 4, 2).$$

Through Equations (2) and (3), the judgment matrix for area E is shown in Table 5.

(ii) Determine the preference of area E over all the other areas. The resulting best-to-others vector would be the following:

$$A_W = (a_{AB}, a_{AC}, a_{AD}, a_{AE}) = (1, 0.5, 0.2, 0.3333).$$

Through Equations (2) and (3), the judgment matrix for area A is shown in Table 6.

(iii) Through Equation (4), the final judgment matrix is shown in Table 7. Standardizing the above matrix by normalization, the areas weights are calculated as follows:

$$\omega_j = (\ 0.0887,\ 0.1244,\ 0.1295,\ 0.3137,\ 0.3437).$$

**Table 5.** Judgment matrix for area E.

| $a_{ij}$ | A | B | C | D | E |
|------|------|------|------|------|--------|
| A | 1 | 0.6 | 0.8 | 0.4 | 0.2 |
| B | 1.67 | 1 | 1.32 | 0.66 | 0.3333 |
| C | 1.25 | 0.75 | 1 | 0.5 | 0.25 |
| D | 2.5 | 1.5 | 2 | 1 | 0.5 |
| E | 5 | 3 | 4 | 2 | 1 |

**Table 6.** Judgment matrix for area A.

| *aij* | A | B | C | D | E |
|---|---|---|---|---|---|
| A | 1 | 1 | 0.5 | 0.2 | 0.3333 |
| B | 1 | 1 | 1 | 0.2 | 0.3333 |
| C | 2 | 1 | 1 | 0.4 | 0.67 |
| D | 5 | 5 | 2.5 | 1 | 1.67 |
| E | 3 | 3 | 1.5 | 0.6 | 1 |

**Table 7.** Final judgment matrix.

| *aij* | A | B | C | D | E |
|---|---|---|---|---|---|
| A | 1.0000 | 0.7746 | 0.6325 | 0.2828 | 0.2569 |
| B | 1.2923 | 1.0000 | 1.1489 | 0.3633 | 0.3300 |
| C | 1.5811 | 0.8660 | 1.0000 | 0.4472 | 0.4093 |
| D | 3.5355 | 2.7386 | 2.2361 | 1.0000 | 0.9138 |
| E | 3.8730 | 3.0000 | 2.4495 | 1.0954 | 1.0000 |

**Step 3.** R software is used to solve the problem in the example on a computer with CPU of 2.3 GHz. Set the maximum number of iterations to 100 times.

The global optimal value is 0.89–0.75, and the running time is about 180 s to 200 s. The results show that this method can produce a feasible optimal scheme. Select a run result as the optimal result. The top five plans are as shown in Table 8.

As shown in Table 8, DMs would better allocate 28 rescuers to area A, 29 rescuers to area B, 22 rescuers to area C, 33 rescuers to area D, and 58 rescuers to area E. In terms of emergency materials, DMs would better allocate 31 kg to area A, 61 kg to area B, 68 kg to area C, 94 kg to area D, and 96 kg to area E. The fitness curve of the run results is shown in Figure 2.

**Table 8.** Run results.

| | X | | | | | Y | | | | | |
|---|---|---|---|---|---|---|---|---|---|---|---|
| | A | B | C | D | E | A | B | C | D | E | *v* |
| 1 | 28 | 29 | 22 | 33 | 58 | 31 | 61 | 68 | 94 | 96 | 0.8885 |
| 2 | 25 | 31 | 28 | 30 | 56 | 34 | 60 | 64 | 93 | 99 | 0.8866 |
| 3 | 25 | 31 | 28 | 32 | 54 | 34 | 60 | 64 | 93 | 99 | 0.8824 |
| 4 | 29 | 30 | 19 | 29 | 63 | 40 | 56 | 71 | 92 | 91 | 0.8677 |
| 5 | 31 | 26 | 22 | 31 | 60 | 31 | 54 | 56 | 103 | 106 | 0.8577 |

Figure 2 shows that the fitness function converges rapidly with iteration. The whole curve shows an upward trend of optimization and tends to the horizontal line of the highest point. The results show that the genetic algorithm has the ability of fast optimization. This method can save decision time and provide advice to decision-makers.

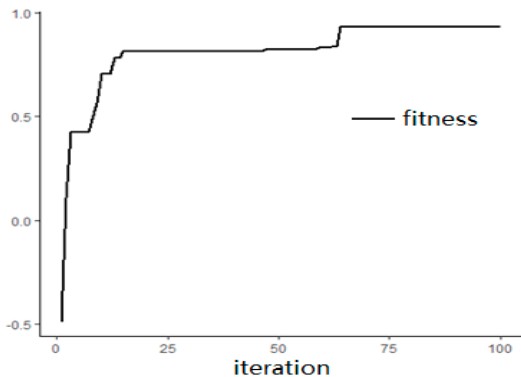

**Figure 2.** The run results.

## 5. Comparative Analyses

In order to reveal the superiority of our proposed method, this section concentrates on comparative analyses with Zhan's method [19] and the standard GA.

We adopt Zhan's method [19] and the standard GA to solve the practical example in Section 4. The computation results are shown in Table 9.

**Table 9.** Comparative results.

| | X | | | | | Y | | | | The Number of Iterations | Average Computation Time |
|---|---|---|---|---|---|---|---|---|---|---|---|
| The improved GA | 28 | 29 | 22 | 33 | 58 | 31 | 61 | 68 | 94 | 96 | 100 | 190 s |
| Zhan's method | 43 | 60 | 67 | 0 | 0 | 84 | 110 | 64 | 50 | 42 | 2200 | 100 s |
| The standard GA | 52 | 50 | 68 | 0 | 0 | 35 | 65 | 88 | 54 | 108 | 100 | 13 s |

Table 9 reveals that the computation results of the same example are different. Compared with Zhan's method and the standard GA, the proposed algorithm has the following advantages. Firstly, Table 9 shows that the computation result of our proposed algorithm is more reasonable than the others. Zhan's method and the standard GA may derive extreme distribution schemes. The reason for this is that they do not take area weights into account, and it disaccords with the views of DMs. Then, although the average computation time of our proposed algorithm is a little longer than the other two methods, in order to achieve better results, the decision-making time is acceptable. The convergence in Zhan's method is difficult, and the proposed algorithm improved this problem. Finally, the algorithm in this paper takes DMs' subjective psychology into consideration, which brings the results closer to reality.

## 6. Conclusions

When a large-scale emergency breaks out, DMs usually face the following problems during the processes of decision making. Firstly, there are many affected areas at the same time. Then, the demand for rescue resources in each affected area is not clear.

In order to solve these problems, this paper studies the allocation of emergency resources in emergency areas with limited resources and fuzzy needs. The main contributions of this paper are as follows:

(1)  On the basis of the analysis of the importance and loss prediction of the emergency area, a weighted method based on the emergency loss levels is constructed to evaluate the expected value of the decision plan.
(2)  On the basis of the analysis of emergency situations under resource constraints, an improved value measurement function of PT is proposed and applied to the genetic algorithm.
(3)  Combining the weighting algorithm of emergency loss level with the genetic algorithm based on the improved prospect value function, an algorithm for the EDM situation of multiple emergency-affected areas with limited resources and fuzzy requirements is proposed.

The main advantages of this algorithm are the following. The emergency loss level weighting algorithm and the genetic algorithm based on the prospect value function are combined with the subjective and objective methods. This method minimizes the interference of a single method and makes EDM more scientific. The weighting algorithm has low complexity and the improved genetic algorithm has high efficiency, which can shorten the time of emergency decision. The number of decision-making schemes can be controlled within acceptable limits to provide project support for direct or re-decision-making.

The algorithm proposed in this paper still has some limitations. In describing regional demands, the use of interval numbers may be not accurate enough to describe the fuzziness of demands; when

the decision-making environment is more complicated, there may be more kinds of demands to think about. Therefore, the emergency decision in this paper can be analyzed by introducing fuzzy theory and multi-requirement hierarchy.

**Author Contributions:** Conceptualization, Writing—original draft preparation, and Writing—review & editing, L.C.; Visualization, Software, G.G.; Project administration, Funding acquisition, Y.W.

**Funding:** This research was funded by the National Nature Science Foundation of China (No. 61773123) and the Humane and Social Science Fund Project of the Education Ministry (No. 14YJC630056).

**Acknowledgments:** The author would like to thank the anonymous reviewers for their constructive comments, which greatly helped to improve this paper.

**Conflicts of Interest:** The author declares no conflict of interest.

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
