# Peer review of "An Improved Genetic Algorithm for Emergency Decision Making under Resource Constraints Based on Prospect Theory"

_algorithms, doi:10.3390/a12020043_

Round 1

Reviewer 1 Report

The article is very interesting and well structured. There are some issues that should be addressed.

-The acronym BWM is not explained what it means.

-Lines 17-19: “Then, an improved GA based on prospect theory is established to solve the

problem of emergency resource allocation between multiple emergency locations with fuzzy

requirements under resource constraints”. The authors do not explain anywhere in the text how fuzzy requirements are incorporated in their proposed methodology.

-In the first paragraph – lines 26-31 there is no literature mentioned.

-Line 84: “The vectors would be AB=…, AW=…”. The authors do not explain what AB, AW

Mean.

-Line 211: the authors refer “Step 4. Repeat steps 3 and 4” probably they mean steps 2 and 3 and not 3 and 4.

-Line 214: the authors refer “Step 5. After the iteration is completed, stop the calculation and output the optimal value”. It is not clear which is the termination criteria for the proposed algorithm. A fixed value of the number of iterations? A fixed value of the fitness function? Or something else?

- It is not clear why the authors present the Judgment matrices only for areas A and E.

-The authors should discuss more about the novelty of their study and compare it with similar studies.

-The results of the Practical Example for EDM with Improved GA should be compared with the standard GA to justify why the proposed algorithm is better than GA.

-The data are real or not. If the data are real the source should be mentioned.

-The limitations of the research are not mentioned.

Author Response

Point 1: -The acronym BWM is not explained what it means.

Response 1: BWM” which is mentioned in the abstract section has been changed into its full name, “best-worst method”. “BWM” which is mentioned in the abstract section means “best-worst method”, and its meaning has been explained (see lines 93-114 in the revised version).

Point 2:-Lines 17-19: “Then, an improved GA based on prospect theory is established to solve the problem of emergency resource allocation between multiple emergency locations with fuzzy requirements under resource constraints”. The authors do not explain anywhere in the text how fuzzy requirements are incorporated in their proposed methodology.

Response 2: “fuzzy requirements” which is mentioned in lines 17-19 has been changed into “uncertain demands”, because the words “uncertain demands” can describe the meaning of this paper more accurately. In the revised version, the explanation of “uncertain demands” has been added in section 3.3 (see lines 192-196).

Point 3:-In the first paragraph – lines 26-31 there is no literature mentioned.

Response 3: Several literatures have been added in the first paragraph (see lines -in the revised version).

Point 4:-Line 84: “The vectors would be AB=…, AW=…”. The authors do not explain what AB, AW Mean.

Response 4: The explanation of AB and AW have been added in section 2.2 (see line 30 in the revised version).

Point 5:-Line 211: the authors refer “Step 4. Repeat steps 3 and 4” probably they mean steps 2 and 3 and not 3 and 4.

Response 5: “Step 4. Repeat steps 3 and 4” in section 3.3. has been changed into “Step 4. Repeat steps 2 and 3” (see line 231 in the revised version)

Point 6:-Line 214: the authors refer “Step 5. After the iteration is completed, stop the calculation and output the optimal value”. It is not clear which is the termination criteria for the proposed algorithm. A fixed value of the number of iterations? A fixed value of the fitness function? Or something else?

Response 6: The termination criteria for the proposed algorithm is a fixed value of the number of iterations. And the number can be set by decision makers. To elaborate on this point, a flowchart has been added in section 3.3 (see Figure 1 in the revised version).

Point 7:- It is not clear why the authors present the Judgment matrices only for areas A and E.

Response 7: In this paper, the best-worst method (BWM) is applied to calculate the area weights. According to the best-worst method, decision makers should decide the best important area and the least important area firstly. And then, in order to get the judgment matrices, decision makers compare the best important area and the least important area with the other areas, respectively. In this paper, area E is the best important area and area A is the least important area. Therefore, the judgment matrices  for areas A and E are only presented. More details about BWM have been added in section 2.2.

Point 8:-The authors should discuss more about the novelty of their study and compare it with similar studies.

Response 8: The discussions about the novelty of our study have been added in section 2.3 (see lines -139-141) and section 5 (see lines 318- 327). The comparison between our proposed approach and other similar study has been added in section 5.

Point 9:-The results of the Practical Example for EDM with Improved GA should be compared with the standard GA to justify why the proposed algorithm is better than GA.

Response 9: The comparison between our proposed improved GA and the standard GA have been added in section 5.

Point 10:-The data are real or not. If the data are real the source should be mentioned.

Response 10: The data in this paper is not real. The algorism in this paper is proposed only for more possibilities about the way on decision making. The researches on emergency respond problem are used to illustrating the calculating procedures with fictional data, such as literature [1], literature [2] and literature [3].  

Point 11:-The limitations of the research are not mentioned.

Response 11: Some limitations of the research are added in the revised version in section 5 (see lines 351- 355).

References

1.      Zhang, J. H.; Li, J.; Liu, Z. P. Multiple-resource and multiple-depot emergency response problem considering secondary disasters. Expert Systems with Applications, 2012, 39, 11066-11071. [CrossRef]

2.      Wang, L.; Zhang, Z. X.; Wang, Y. M. A prospect theory-based interval dynamic reference point method for emergency decision making. Expert Systems with Applications, 2015, 42, 9379-9388. [CrossRef]

3.      Zhang, Z. X.; Wang, L.; Rosa M. Rodríguez; Wang, Y. M.; Martínez, L. A hesitant group emergency decision making method based on prospect theory. Complex & Intelligent Systems, 2017, 3, 177–187. [CrossRef]

Reviewer 2 Report

The topic is of interest to safety professionals as well as academic. However, the paper is poorly written and substantial changes need to be made in order for the paper to be considered for publication. Below are some comments to help Authors improve their paper.

The abbreviations used in the article must be clearly identified in the main text of the article. It is a mistake to use abbreviations without explanation. This applies to abbreviations among others explanation only in the summary, e.g. "DM", "EDM", "BWM".

The text format is incorrect, unreadable (no justify text on a page). This should be improved.

The literature review is very weak. For example: the Genetic Algorithm is known for many scientific publications, but the Authors do not refer to any publications in point 2.3 "Genetic Algorithm". In the text the description of the algorithm is not a new description of existed algorithm.

Although the literature review refers to the most recent publications (2010-2018), there is one publication from 1987 used to the description of point 2.2 "Best-worst Method". Is it means that since 1987 this method has not been made used in any research?

The general comment: the literature review needs to be improved. Besides, the literature review section should be written in a way to identify the gap in knowledge and explain why it is paramount to bridge the identified gap. After then, the study objectives should be stated in a direct manner.

Section 3 "An Algorithm for EDM with GA". The authors refer to the definition, the prospect theory, but again do not give any literature. This is an unacceptable practice used by the Authors to define concepts and definitions throughout the article.

Section 4 "Practical Example for EDM with Improved GA"
The authors write that: "In recent years, with the emergence of typhoid fever, cholera, malaria and other diseases, floods, earthquakes and typhoons have become more and more frequent around the world".
Please indicate the source of data on the basis of which it is possible to say that these phenomena are "more frequent around the world".

Please check that all formulas, values, numbers are correctly entered in the text. Sometimes the values are illegible, such as lines 143, 144, 192, 243, 268

Please check the language of the article and correct the typos, e.g. line 300 "the geneti4 algorithm".

Author Response

Point 1: The abbreviations used in the article must be clearly identified in the main text of the article. It is a mistake to use abbreviations without explanation. This applies to abbreviations among others explanation only in the summary, e.g. "DM", "EDM", "BWM".

Response 1: The explanations of all the abbreviations have been added in abstract and the main text of the article.

Point 2: The text format is incorrect, unreadable (no justify text on a page). This should be improved.

Response 2: The text format has been reformatted in the revised version.

Point 3: The literature review is very weak. For example: the Genetic Algorithm is known for many scientific publications, but the Authors do not refer to any publications in point 2.3 "Genetic Algorithm". In the text the description of the algorithm is not a new description of existed algorithm.

Response 3: Several literatures about the Genetic Algorithm have been added into section 2.3 (see lines 135-139).

Point 4: Although the literature review refers to the most recent publications (2010-2018), there is one publication from 1987 used to the description of point 2.2 "Best-worst Method". Is it means that since 1987 this method has not been made used in any research?

Response 4: The best-worst method (BWM) mentioned in this paper was proposed by Jafar Rezaei in 2015 (see Reference 25 in line 417). Several literatures about BWM have been added in section 2.2 to better introduce the method. More details about BWM have been added in section 2.2.

Point 5: The general comment: the literature review needs to be improved. Besides, the literature review section should be written in a way to identify the gap in knowledge and explain why it is paramount to bridge the identified gap. After then, the study objectives should be stated in a direct manner.

Response 5: The literature review has been improved in the revised version. Several literatures have been added in the revised version.

Point 6: Section 3 "An Algorithm for EDM with GA". The authors refer to the definition, the prospect theory, but again do not give any literature. This is an unacceptable practice used by the Authors to define concepts and definitions throughout the article.

Response 6: We have corrected the error about defining concepts and definitions by the Authors. Several literatures about the prospect theory have been added in section 2.1 (see lines 85-90).

Point 7: Section 4 "Practical Example for EDM with Improved GA"

The authors write that: "In recent years, with the emergence of typhoid fever, cholera, malaria and other diseases, floods, earthquakes and typhoons have become more and more frequent around the world".

Please indicate the source of data on the basis of which it is possible to say that these phenomena are "more frequent around the world".

Response 7: "In recent years, with the emergence of typhoid fever, cholera, malaria and other diseases, floods, earthquakes and typhoons have become more and more frequent around the world" mentioned in this paper has been deleted in revised version.

Point 8: Please check that all formulas, values, numbers are correctly entered in the text. Sometimes the values are illegible, such as lines 143, 144, 192, 243, 268

Response 8: The formulas and values have been enlarged. Some spaces have been put between values for making them easy to read.

Point 9: Please check the language of the article and correct the typos, e.g. line 300 "the geneti4 algorithm".

Response 9: The language of the article has been checked. And the typos in this paper have been corrected.

Round 2

Reviewer 1 Report

The manuscript has been improved.

-Lines 31-32 “As a consequence, the topic about emergency response has touched off the heated discussions recently” this phrase should be expressed in another way. This is not an appropriate expression for a scientific article.

-Line 31 “Tale 1” should be corrected to “Table 1”

-Line 115 “In this paper, this method was employed in order to weight the area” the phrase “to weight the area” should be expressed in another way. The area cannot be weighted.

-Why the mutation is not mentioned in the proposed algorithm? The improved GA has no mutation stage?

Author Response

Point 1:-Lines 31-32 “As a consequence, the topic about emergency response has touched off the heated discussions recently” this phrase should be expressed in another way. This is not an appropriate expression for a scientific article. 

Response 1: “As a consequence, the topic about emergency response has touched off the heated discussions recently” has been changed into “the topic about emergency response has attracted some scholars’ attention recently” in line 27.

Point 2:-Line 31 “Tale 1” should be corrected to “Table 1”

Response 2: “Tale 1” in line 31 has been corrected to “Table 1” in line 27 in the revised version.

Point 3:-Line 115 “In this paper, this method was employed in order to weight the area” the phrase “to weight the area” should be expressed in another way. The area cannot be weighted.

Response 3: The phrase “to weight the area” has been changed into “to calculate the priority weighting of the areas” in line 111 in the revised version.

Point 4:-Why the mutation is not mentioned in the proposed algorithm? The improved GA has no mutation stage?

Response 4: The mutation step has been added in section 3.3 (see line 242) and Figure 1 in the revised version.

Reviewer 2 Report

The Authors have revised all the comments from the 1st round (those I mentioned).

Anyway, except there was very difficult process of second verifying article in "track changes" used by the Authors, I have no more comments.

Author Response

Point 1: Anyway, except there was very difficult process of second verifying article in "track changes" used by the Authors, I have no more comments.

Response 1: The Word file has been changed into revisions mode in the revised version.